# Immunoproteasome Inhibition Impairs Differentiation but Not Survival of T Helper 17 Cells

**DOI:** 10.3390/cells14100689

**Published:** 2025-05-10

**Authors:** Franziska Oliveri, Dennis Mink, Tony Muchamuel, Michael Basler

**Affiliations:** 1Biotechnology Institute Thurgau at the University of Konstanz, 8280 Kreuzlingen, Switzerland; 2Division of Immunology, Department of Biology, University of Konstanz, 78457 Konstanz, Germany; 3Department of Research, Kezar Life Sciences, Inc., South San Francisco, CA 94080, USA

**Keywords:** immunoproteasome, Th17 cells, Tregs, colitis, acute inflammation

## Abstract

Autoimmune and inflammatory diseases are characterized by aberrant immune responses. The immunoproteasome was proposed as a target for such Th cell-mediated diseases due to its role in the activation, differentiation and function of T cells. Even though many studies demonstrated reductions in Th17 cells upon immunoproteasome inhibition, it is still unclear if the differentiation or survival of these cells is affected. Therefore, this study used DSS-induced colitis and house dust mite airway inflammation mouse models to investigate the effect of immunoproteasome inhibition on Th17 cells and Tregs at different time points. Th17 cells were almost abolished when immunoproteasome inhibition was applied continuously in DSS-induced colitis. In contrast, immunoproteasome inhibition did not decrease levels of already differentiated Th17 cells and did not enhance Treg induction. Dendritic cells were barely affected by immunoproteasome inhibition. Moreover, immunoproteasome inhibition reduced T cell activation in vitro and in vivo, suggesting impaired activation as the underlying mechanism for reduced Th17 differentiation. In conclusion, immunoproteasome inhibition reduces Th17 differentiation by impairing the activation of naïve T cells, but it does not affect the survival of already-differentiated Th17 cells and Tregs.

## 1. Introduction

Th17 cells are critical players in anti-fungal and anti-bacterial immune responses, especially in mucosal tissues [1]. After their discovery almost 20 years ago [2], Th17 cells have gained attention for their involvement in driving autoimmune and chronic inflammatory diseases such as inflammatory bowel disease (IBD), rheumatoid arthritis, multiple sclerosis (MS) and systemic lupus erythematosus (SLE) (summarized in [3]). In contrast, a main role of immunosuppressive regulatory T cells (Tregs) is the prevention of autoimmunity [4]. Even though several Treg and Th17 subpopulations exist in respect to cytokine profile, origin and pro-/anti-inflammatory functions, it is generally accepted that the Th17/Treg balance is crucial for the maintenance of self-tolerance, and its disruption leads to chronic inflammation and autoimmunity [5].

In the last few years, the immunoproteasome was proposed as a target for T helper cell-mediated diseases such as autoimmune disorders and chronic inflammation due to its role in the activation, differentiation and function of T cells, which goes beyond its important role in antigen presentation [6,7,8,9,10,11,12,13]. The immunoproteasome is a variant of the 26S proteasome, which contains low molecular mass polypeptide (LMP)2 (β1i), multicatalytic endopeptidase complex-like (MECL)-1 (β2i) and LMP7 (β5i) instead of the standard catalytically active β-subunits (β1c, β2c and β5c). It is constitutively expressed in cells of hematopoietic origin [14] and can be induced in other cell types by interferon-γ (IFN-γ) [15,16,17,18]. Due to this specific expression of the immunoproteasome in immune cells, immunoproteasome-specific inhibitors were developed as an alternative to unspecific proteasome inhibitors such as bortezomib and carfilzomib, with a better safety profile [8].

We recently demonstrated the strong expression of the immunoproteasome compared to standard proteasome in different T helper cell subsets, including Th17 and Tregs [19], suggesting a crucial role in these cells. In line with this, previous work by our group and others has indicated that immunoproteasome inhibition affects T helper cell commitment, as shown for Th1, Th2 and especially Th17 cells. In particular, Th17 polarization was impaired by immunoproteasome inhibition in vitro [7,13,20]. Moreover, various studies demonstrated beneficial effects of immunoproteasome inhibition in mouse models of autoimmune and inflammatory diseases such as IBD [7,9], rheumatoid arthritis [8], experimental autoimmune encephalomyelitis (EAE) [10], Hashimoto thyroiditis [21], SLE [22], experimental autoimmune neuritis [20], myasthenia gravis [23], Sjörgen’s syndrome [11], polymyositis [24], psoriasis [25] and allergic airway inflammation [19]. In several of these studies, Th17 cells were reduced by immunoproteasome inhibition, e.g., showing a reduction in Th17 cells in the lamina propria of DSS-induced colitis mice [7] and in lymph nodes of EAE mice [10].

Even though these studies clearly demonstrate a role of the immunoproteasome in T helper cells, it is still unknown whether the initial differentiation and/or subsequent survival of these cells is affected by immunoproteasome inhibition. In order to address this question, we used mouse models of DSS-induced colitis and airway inflammation to study the effect of immunoproteasome inhibition on Th17 cells at different time points. Moreover, we investigated Tregs in the DSS setting, since it was previously suggested that immunoproteasome inhibition affects the Th17/Treg balance. We found that immunoproteasome inhibition interfered with initial Th17 differentiation, but did not affect the survival of pre-existing Th17 cells.

## 2. Materials and Methods

### 2.1. Mice

IL-17A-GFP (C57BL/6-*Il17a^tm1Bcgen^*/J; stock #018472) mice were purchased from The Jackson Laboratories. FoxP3-GFP (C57BL/6-*Foxp3^tm1Kuch^*) [26] mice were kindly provided by Nicole Joller (University of Zurich, Institute of Experimental Immunology, Zurich, Switzerland). Of note, to prevent mouse strain-specific variation and to reduce the number of mice according to the 3R principle for animal experimentation, we used IL-17A-GFP mice, although the reporter construct was not used for analysis in certain experiments. GATIR (*Gata3^tm1.1Hjf^*) [27] mice were kindly provided by Hans Joerg Fehling (Institute of Immunology, University Clinics Ulm, Ulm, Germany). Male and female mice were used at 8–14 weeks of age. All mice were kept in a specific pathogen-free environment with ad libitum access to food and water and a 12/12 h light–dark cycle. Animal experiments were approved by the reviewing board of the Regierungspräsidium Freiburg according to the local ethic guidelines (G-19/43; G-21/002; T-21/03TFA; T-24/02TFA).

### 2.2. DSS-Induced Colitis

IL-17A-GFP and FoxP3-GFP mice were exposed to dextran sulfate sodium (DSS; m.w. 36,000–50,000; MP Biomedicals, Eschwege, Germany) by addition of 2% (*w*/*v*) to the drinking water for the induction of colitis. After 5 days, mice received normal drinking water. Body weight was monitored daily throughout the experiment. The induction of Th17 cells and Tregs was analyzed by flow cytometry at different time points.

### 2.3. House Dust Mite Induced Airway Inflammation (HDM-AI)

House dust mite (HDM) extract was used to induce airway inflammation (AI) in GATIR mice. Anesthetized mice received intranasal applications of 50 µg D. pteronyssinus extract (Citeq Biologicals, Groningen, The Netherlands) in PBS on day 0, 7, 14 and 21 as described before [19]. Infiltration of Th17 cells into the lung was analyzed on day 21 and 23.

### 2.4. Immunoproteasome Inhibition by ONX 0914

ONX 0914 [8] (Kezar Life Sciences, South San Francisco, CA, USA) was used for the in vivo irreversible inhibition of the immunoproteasome. It was formulated in a solution of 10% sulfobutylether-β-cyclodextrin (*w*/*v*; Captisol, San Diego, CA, USA) and 10 mM sodium citrate (pH 6) and was used at a dose of 10 mg/kg, administered subcutaneously into the loose skin at the neck. Control mice received the same volume of vehicle at the same time points.

For the analysis of the effect of immunoproteasome inhibition on the differentiation of Th17 cells, mice were treated with ONX 0914 every other day starting from day 0 (“prophylactic”). To investigate Th17 survival, ONX 0914 treatment started after the DSS cycle on day 6 (“therapeutic”).

In HDM-AI experiments, mice received a single dose of ONX 0914 or vehicle on day 21 one hour before the intranasal challenge.

ONX 0914 was dissolved in dimethyl sulfoxide (DMSO) and used at 200 nM and 300 nM for in vitro experiments.

### 2.5. Organ Preparation

Mice were sacrificed by CO_2_ asphyxia or cervical dislocation. Spleen, iliac and mesenteric lymph nodes were collected in FACS buffer (PBS + 2% FCS + 2 mM EDTA) and single cell suspensions were prepared using 70 µm and 100 µm nylon meshes (Greiner Bio-one, Frickenhausen, Germany).

The colon was dissected and the length was measured. Lamina propria lymphocytes were isolated using the Lamina Propria Dissociation Kit according to the manufacturer’s instructions (Miltenyi Biotech, Bergisch Gladbach, Germany). In brief, colons were flushed with Hank’s balanced salt solution (HBSS) without Ca^2+^ and Mg^2+^ to remove feces, and cut open using Metzenbaum scissors. They were cut into small pieces and transferred into pre-digestion solution (HBSS with Ca^2+^ and Mg^2+^, 5 mM EDTA, 5% FCS, 1 mM dithiothreitol). Samples were subjected to three incubation rounds in pre-digestion solution at 37 °C for 20 min with mixing and filtration through 100 µm MACS SmartStrainers in between to remove intraepithelial lymphocytes. Remaining tissue pieces were transferred into MACS C tubes containing digestion solution (HBSS with Ca^2+^ and Mg^2+^, 5% FCS, enzyme mix) and dissociated into single cell suspensions using the gentleMACS Octo Dissociator (Miltenyi Biotech, Bergisch Gladbach, Germany; program—37 °C_m_LPDK_1). Single cell suspensions were again filtered (100 µM) and collected by centrifugation at 300× *g* for 10 min.

Lungs from HDM-AI mice were dissected after bronchoalveolar lavage and washed once with PBS. Single cell suspensions were obtained using the Lung Dissociation Kit (Miltenyi Biotech) as described before [19].

Samples were resuspended in FACS buffer and analyzed by flow cytometry.

### 2.6. Flow Cytometric Analysis

Single cell suspensions were used for flow cytometry. First, Fc-receptor blocking was performed using the 2.4G2 antibody (obtained from 2.4G2 hybridoma cells) for 20 min on ice. Staining with the fixable viability dye 780 (FVS780) (BD Biosciences, Heidelberg, Germany) was included according to the manufacturer’s instructions for samples that were subsequently fixed with paraformaldehyde. Cells were washed once with FACS buffer and stained with different antibody cocktails (list of antibodies in Appendix A) for 30 min on ice followed by two washing steps. Due to the extensive processing and measurement time, samples from DSS-induced experiments where fixed with 4% paraformaldehyde for 5 min on ice. For cells without fixation, staining with Sytox Blue/Red (1:100,000 and 1:50,000, Fisher Scientific, Schwerte, Germany) was included for live/dead cell discrimination. All samples were measured on LSRFortessa, FACSVerse, FACSLyric (all BD Biosciences, Heidelberg, Germany) or CytoFLEX LX™ (Beckman Coulter, Krefeld, Germany). Analyses were performed according to previously described guidelines for flow cytometry [28]. Flow cytometry data were analyzed with FlowJo v10.8.1 (BD Biosciences). Representative gating schemes are depicted in Appendix A.

### 2.7. Intracellular Cytokine Staining

For the analysis of Th17 cells and Tregs in HDM experiments, single cell suspensions from lungs were seeded into a 96-well plate in RPMI containing 10% FCS, 1% penicillin/streptomycin, and 50 µM β-mercaptoethanol. T cells were restimulated using 25 ng/mL phorbol 12-myristate 13-acetate, 500 ng/mL ionomycin and 10 µg/mL brefeldin A (all Merck, Darmstadt, Germany) for 5 h at 37 °C, 5% CO_2_. Fc-receptor blocking, FVS780 and surface staining were performed as described above. Cells were fixed with 4% paraformaldehyde for 5 min on ice and subsequently permeabilizedusing PBS containing 0.1% saponin, 2% FCS, 2 mM EDTA and 2 mM NaN_3_ to allow staining for IL-17 and FoxP3. Samples were measured on LSRFortessa (BD Biosciences, Heidelberg, Germany). Flow cytometry data were analyzed with FlowJo v10 (BD Biosciences, Heidelberg, Germany). Gating schemes are depicted in Appendix A and Figure 1.

### 2.8. In Vitro T Cell Activation

CD4^+^ T cells were isolated from spleens of IL-17A-GFP or FoxP3-GFP using CD4^+^ Microbeads mouse (Miltenyi Biotech, Bergisch Gladbach, Germany) according to the manufacturer’s instructions. Here, 5 × 10^5^ T cells/well were seeded into a 96-well plate coated with 5 µg/mL anti-CD3 and 5 µg/mL anti-CD28 (Biolegend, Amsterdam, The Netherlands) in RPMI containing 10% FCS, 1% penicillin/streptomycin, and 50 µM β-mercaptoethanol. Then, 200 nM and 300 nM ONX 0914 or vehicle (DMSO) was added, starting from the beginning. Samples were incubated at 37 °C, 5% CO_2_, and analyzed by flow cytometry after 12 and 24 h. Surface staining was performed as described above.

### 2.9. In Vitro Th17 Differentiation

Single cell suspensions from spleens of IL-17A-GFP mice were sorted using CD4 (L3T4) MicroBeads mouse (Milteny Biotech, Bergisch Gladbach, Germany) according to the manufacturer’s instructions; 0.25 × 10^6^ T-cells/well were seeded into a 96-well plate coated with 5 µg/mL anti-CD3 and 5 µg/mL anti-CD28 in RPMI containing 10% FCS, 1% penicillin/streptomycin, and 50 µM β-mercaptoethanol. For polarization towards Th17 cells, T cells were stimulated using the cytokines IL-6 (20 ng/mL), IL-23 (10 ng/mL), IL-1β (10 ng/mL) and TGF-β1 (2 ng/mL), as well as blocking antibodies anti-IL-4 (10 µg/mL), anti-IFN-γ (10 µg/mL), and anti-IL-2 (10 µg/mL) (CytoBox Th17, mouse, Milteny Biotech, Bergisch Gladbach, Germany). Cells were incubated at 37 °C, 5% CO_2_, for 4 days before treatment with 100 nM, 200 nM, and 300 nM ONX 0914 or vehicle (DMSO). After overnight incubation for 16 h, cells were stained with PE anti-mouse CD4, Annexin-V APC (1:100, Biolegend, Amsterdam, The Netherlands) and Sytox Blue (1:100,000, Fisher Scientific, Schwerte, Germany).

### 2.10. Statistics

Data were evaluated using GraphPad Prism Version 9/10 and are shown as mean ± standard deviation (SD). Details regarding sample size and replication are listed in the figure legends. Kolmogorov–Smirnov and Shapiro–Wilk tests were performed to assess normality. For normally distributed data, 1- or 2-way-ANOVA or Student’s *t*-test was performed to determine the statistical significance of differences. Holm–Sidak (body weight) or Turkey’s test was used for multiple comparison of numerous different groups. The Kruskal–Wallis or Mann–Whitney test was used for non-normally distributed data. Statistical significance was achieved when *p* < 0.05; * *p* < 0.05, ** *p* < 0.01, *** *p* < 0.001, and **** *p* < 0.0001.

## 3. Results

### 3.1. Impaired Induction of Th17 Response upon Immunoproteasome Inhibition in DSS-Induced Colitis

Previous work indicates a strong impact of immunoproteasome inhibition on T helper cells, especially Th17 cells, in vitro [7,8] and in vivo [7,10,25]. However, it remained unclear whether their differentiation or survival is affected by immunoproteasome inhibition.

To evaluate the differentiation of Th17 cells in vivo, we used the dextran sulfate sodium (DSS)-induced colitis model in IL-17A-GFP reporter mice. This model is characterized by a strong Th17 response, which was reduced by immunoproteasome inhibition in previous studies performed by our group [7,9]. Since this study aims at further investigating the effect of immunoproteasome inhibition at different stages of Th17 differentiation and survival, this model offers the ideal basis for this due to the defined and well-characterized time frame. After the initial induction of inflammation and the Th17 response, the acute inflammation diminishes as soon as DSS is removed from the drinking water. Therefore, the application of immunoproteasome inhibition at a later time point allows for studying the effect on differentiated Th17 cells without the continuous induction of new Th17 cells.

First, 10 mg/kg ONX 0914 or vehicle was administered subcutaneously starting from the beginning of colitis induction in a “prophylactic” setting. In line with previous studies [7,9], immunoproteasome inhibition by ONX 0914 reduced body weight loss and colon shortening, indicating a reduction in colitis-induced symptoms (Figure 1A–C).

**Figure 1 cells-14-00689-f001:**
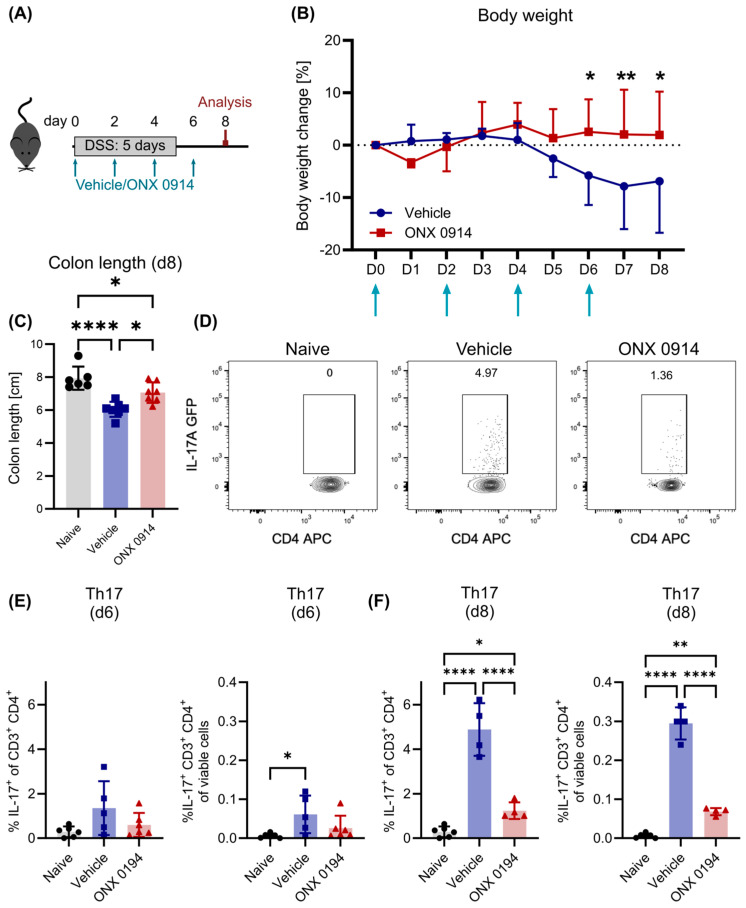
Immunoproteasome inhibition reduces symptoms of DSS-induced colitis. Here, 2% DSS was added to the drinking water of IL-17A-GFP mice for 5 days, followed by normal drinking water. Simultaneously, vehicle or ONX 0914 (10 mg/kg) was administered subcutaneously every other day and mice were sacrificed on day 8, as depicted in (**A**). (**B**) Body weight was monitored daily. Percent weight loss relative to the weight on day 0 (*y*-axis) is plotted versus time in days (*x*-axis). Vehicle or ONX 0914 treatment time points are indicated with arrows. (**C**) Mice were sacrificed on day 8 and the colon length was measured after dissection (caudal of the appendix). Flow cytometric analysis was performed on dissected and mechanically/enzymatically dissociated tissues. (**D**) Gating scheme to analyze Th17 cells. After exclusion of doublets and dead cells (gating strategy described in Supplementary Fig. S1), cells were identified as CD3^+^ CD4^+^ and subsequently analyzed for the expression of IL-17A-GFP. Graph shows a sample of the lamina propria on day 8. (**E**,**F**) Frequency of Th17 cells among CD4^+^ T cells (left) and among all viable cells (right) on day 6 (**E**) and day 8 (**F**) in the lamina propria. Graphs show pooled data of 2–3 independent experiments. (**B**) Vehicle/ONX 0914: *n* = 7. (**C**) Naïve: *n* = 6, vehicle/ONX 0914: *n* = 7. (**E**,**F**) *n* = 4–6. Data are shown as mean (**B**) + SD or (**C**–**F**) ± SD. (**C**,**F**) Each point represents an individual mouse. * *p* < 0.05, ** *p* < 0.01, **** *p* < 0.0001 ((**B**) 2-way ANOVA, (**C**,**F**) 1-way ANOVA or Kruskal–Wallis test).

In order to analyze the effect of immunoproteasome inhibition in more detail, the induction of Th17 cells was measured on day 6 and day 8. Flow cytometric analysis revealed that Th17 cells (defined as CD3^+^ CD4^+^ IL-17A GFP^+^, Figure 1D and Appendix A) were strongly induced in the lamina propria of the colon (Figure 1D–F, whereas few Th17 cells could be detected in lymph nodes (mesenteric and inguinal), the spleen, and Peyer’s patches of the small intestine (Appendix A). In the lamina propria, the frequency of Th17 cells increased among all viable cells (Figure 1E,F right) but also among CD4^+^ T cells (Figure 1E,F left), indicating that the observed increase was not only a result of a general increase in infiltrating CD4^+^ T cells. Furthermore, the frequency of Th17 cells increased from day 6 to day 8, which reflects the induction of the adaptive immune response requiring several days. Strikingly, the frequency of Th17 cells was decreased when mice were treated with ONX 0914, especially on day 8 post colitis induction (Figure 1E,F), in line with previous results [7]. Importantly, this was not caused by a general decrease in CD4^+^ T cells in the lamina propria, since the frequency of all CD4^+^ T cells did not change with ONX 0914 treatment (Appendix A) but the frequency of Th17 cells among CD4^+^ T cells decreased significantly.

### 3.2. Immunoproteasome Inhibition Does Not Reduce Differentiated Th17 Cells in DSS-Induced Colitis

Second, we aimed to assess the effect of immunoproteasome inhibition on already differentiated Th17 cells. Therefore, colitis was induced first by DSS administration. ONX 0914 treatment started in a “therapeutic setup” after induction of the Th17 response on day 6 (Figure 2A), a time point at which the induction of newly differentiated Th17 cells can easily be observed (Figure 1E). As expected, mice started recovering weight after day 8 (Figure 2B). On day 12, vehicle- and ONX 0914-treated mice had recovered their original weight and colon length similarly (Figure 2A–C). These results indicate that immunoproteasome inhibition in a therapeutic setup has no positive effect on the recovery of DSS-induced colitis.

On day 10, the frequency of Th17 cells in the lamina propria was similar to that on day 8 (Figure 2D, vehicle group), and it slightly declined on day 12 (Figure 2E). CD4^+^ T cells were more abundant among lamina propria lymphocytes (LPL) on day 10 than on day 8 (Appendix A). Surprisingly, no differences between the two treatment groups could be observed (Figure 2D–F), indicating that immunoproteasome inhibition did not affect the survival of these cells.

### 3.3. No Effect of Immunoproteasome Inhibition on Differentiated Th17 Cells in HDM-AI

To rule out a disease model-dependent effect on Th17 cell differentiation, we set out to confirm our results in a mouse model for asthma. Asthma is a very heterogeneous disease with many different phenotypes besides the most common Th2 cell-meditated eosinophilic form [29]. Since Th17 cells have recently been reported to play a major role in these other forms of asthma [30], we used the model of house dust mite (HDM)-induced acute airway inflammation (AI) to analyze Th17 cells (Figure 3A). In contrast to the dependency of the DSS model on innate immunity, in the HDM-AI model, the disease is induced by an adaptive immune response. This allowed us to validate the effect of immunoproteasome inhibition under two different conditions.

Using this model, we could show that immunoproteasome inhibition reduces the Th2 response by a single application of ONX 0914 before re-challenge with the allergen HDM [19]. Here, we can show now that HDM-AI is not only characterized by a Th2 response, but also, Th17 cells were strongly induced on day 21 after three HDM applications (Figure 3B,C), in line with the literature [31,32]. This population was stable two days later upon re-challenge with the HDM allergen for a fourth time. Like the DSS model, this allowed us to study the effect of immunoproteasome inhibition on differentiated Th17 cells at a defined time point without the induction of new Th17 cells. Interestingly, Th17 cells were not affected by a single dose of ONX 0914 before the re-challenge (Figure 3C,D), confirming the results from DSS-induced colitis (Figure 2) and showing that the survival of existing Th17 cells is not affected by immunoproteasome inhibition. Taken together, our results show that immunoproteasome inhibition does not affect the survival of Th17, at least in short-term experiments.

### 3.4. Immunoproteasome Inhibition Does Not Enhance the Induction of Tregs in DSS-Induced Colitis

It was previously suggested that the reduction in Th17 cells upon immunoproteasome inhibition is accompanied, or even results from, an increase in Tregs, since a dysregulation of their balance is often found in autoimmune diseases [7]. Because they are important regulators in the modulation of immune responses to prevent excessive inflammation [33], they are also induced with a similar kinetic to Th17 cells in DSS-induced colitis. Accordingly, the frequency of Tregs in the lamina propria of DSS-treated mice was increased on day 6 compared to naïve mice (Figure 4B,C), and was slightly enhanced on day 8 (Figure 4D). In contrast to the effect on Th17 cells, there was no difference detectable between the two treatment groups, indicating that immunoproteasome inhibition does not affect the development of Tregs. In line with these results, the application of ONX 0914 after DSS treatment on day 6 and day 8 did not affect the frequency of Tregs on day 10 in the lamina propria (Appendix A). Since a previously published study by Kalim et al. investigated Tregs not in the lamina propria but in mesenteric lymph nodes [7], we also analyzed the lymph nodes in our study. There was no difference from naïve mice on day 6 (Appendix A), but Tregs were induced in the vehicle group on day 8 (Figure 4E). Interestingly, whereas Tregs were induced in the vehicle treated group in some mice, ONX 0914 treatment prevented Tregs induction in the lymph nodes compared to the lamina propria. Moreover, we analyzed Tregs in HDM-AI, as described in the previous section. Immunoproteasome inhibition did not affect Treg frequencies either in the lung or in the spleen (Appendix A). Taken together, Tregs do not require functional immunoproteasome for their development, and enhanced Treg differentiation in our in vivo setups could not be detected.

### 3.5. Immunoproteasome Inhibition Slightly Reduces Dendritic Cells in Mesenteric Lymph Nodes

Dendritic cells (DCs) play a major role in the induction of Th17 cells in the intestine [34], and were reported to be affected by immunoproteasome inhibition [22,35,36,37,38]. Hence, an altered DC population in immunoproteasome inhibitor-treated mice might contribute to the reduced induction of Th17 cells in DSS-induced colitis (Figure 1). Therefore, the frequency of CD11c^+^ cells was analyzed in the lamina propria of the colon and in mesenteric lymph nodes. CD11c^+^ cells were slightly reduced in the lamina propria of ONX 0914-treated mice on day 6 and day 8, even though the data display high variation and the differences are not statistically significant (Figure 5A,C,E). Mesenteric lymph nodes are the major site for DC/T cell interaction [39], and accordingly, a robust population of CD11c^+^ cells could be detected there (Figure 5B). The variance on day 6 was very high, making a definite conclusion difficult, but interestingly, the CD11c^+^ population was markedly reduced on day 8 (Figure 5F), thus suggesting that DCs might be affected by immunoproteasome inhibition. Taken together, CD11c^+^ population in the lamina propria and in mesenteric lymph nodes was only slightly reduced in mice with DSS-induced colitis, and, in our opinion, is not the driving factor for the observed reduced induction in Th17 cells.

### 3.6. Immunoproteasome Inhibition Reduces T Cell Activation In Vivo and In Vitro

Our group previously showed that immunoproteasome inhibition impairs the activation of T cells [6,19]. Since the activation of T cells is a critical step in their differentiation process, we analyzed the activation status of CD4^+^ T cells in DSS-induced colitis by flow cytometry. Since Treg levels are increased in DSS-induced colitis on day 8, but we wanted to focus on conventional T helper cells, we made use of the FoxP3-GFP reporter mice to analyze all non-Treg cells by gating on FoxP3-negative CD4^+^ T cells. Interestingly, on day 8 post-induction, we found a reduction in the activated CD44^hi^ FoxP3^-^ CD4^+^ T cell population as well as a reduced overall expression level of CD44 on FoxP3^-^ CD4^+^ T cells in ONX 0914-treated mice (Figure 6A). Because T cell activation is an early event during differentiation, we investigated the activation of T cells in more detail at an earlier time point (day 4). The expression levels of both activation markers CD44 and CD69 were reduced on CD4^+^ T cells in the spleen (Figure 6B–D), suggesting that impaired early T cell activation might be responsible for the observed impairment of Th17 differentiation.

The observed reduction in activated T cells could result from different processes. On the one hand, immunoproteasome inhibition could directly impair activation mechanisms and thus block the transition from naïve to an activated state. On the other hand, activated cells could be dying as a result of the induction of apoptosis upon immunoproteasome inhibition, thus also resulting in lower levels of activated T cells. Since we do not know the exact kinetics of this process in vivo, we isolated CD4^+^ T cells from murine spleens and activated them in vitro with plate-bound anti-CD3/CD28 antibodies. After 24 h, the expression levels of activation markers CD44, CD223 (LAG3) and PD-1 were increased on stimulated live CD4^+^ T cells and, similar to in vivo results, levels were reduced in the presence of ONX 0914 (Figure 6E,F). As expected, the stimulation of T cells led to a downregulation of CD62L, a common marker of naïve T cells, as was also seen on ONX 0914-treated cells (Figure 6H). Even though differences in activation between vehicle- and ONX 0914-treated samples could be observed, all living CD4^+^ T cells clearly showed a loss of the naïve phenotype, indicating that a complete blockade in the transition from a naïve to an activated state is not the underlying mechanism for the observed reduction in activated T cells in the presence of immunoproteasome inhibition. Moreover, it rather seems like a generally impaired activation, which could, for example, be explained by the dysfunctionality of these cells. The activation of T cells slightly increased the viability of CD4^+^ T cells compared to non-activated cells 12 h post stimulation (Figure 6I). Since the viability of ONX 0914 is only slightly reduced after 12 h, it seems that ONX 0914 does not simply kill the naïve T cells. In contrast, 24 h post T cell activation, the viability of ONX 0914-treated T cells decreased to levels similar to unstimulated vehicle treated cells. We then sought to investigate whether Th17 cells are more susceptible to immunoproteasome inhibition than other CD4^+^ cells. CD4^+^ T cells from IL-17A-GFP reporter mice were in vitro differentiated to Th17 cells and treated with ONX 0914 for 16 h. Analysis of apoptosis by flow cytometry revealed a mild increase in apoptotic cells at 300 nM ONX 0914 (Appendix A). However, Th17 cells did not exhibit higher susceptibility to immunoproteasome inhibition compared to IL-17A-negative T helper cells. Taken together, although T cells activated in the presence of immunoproteasome inhibition initially make the transition from a naïve to an activated stage, they are improperly activated, partially leading to cell death.

## 4. Discussion

Several studies using pre-clinical mouse models of (auto-) inflammatory diseases have shown that immunoproteasome inhibition has great therapeutic potential, for example in SLE, EAE and IBD [7,9,10,22]. Previous work indicates a strong influence of immunoproteasome inhibition on T helper cells, especially Th17 cells, which was demonstrated in vitro [7,8] and in vivo [7,10,25]. In vivo studies reported a reduction in Th17 cells upon immunoproteasome inhibition in several mouse models, but it is not clear whether the differentiation or survival of Th17 cells was affected. Furthermore, it was suggested that an increase in Tregs could accompany the reduction in Th17 cells, because autoimmune diseases often present with a dysregulated Th17/Treg-balance [33]. Until now, there have been few studies investigating the effect of immunoproteasome inhibition on Tregs, yielding inconsistent results. To shed light on the effect of immunoproteasome inhibition on those two cell types (Th17 and Treg), we used the DSS-induced colitis model to induce Th17 cells and Tregs, and applied the immunoproteasome inhibitor ONX 0914 at different time points. This model allowed us to study Th17 cells and Tregs independently but in the same setting, since this colitis animal model is not Th17- or Treg-driven, but these cells are rather induced as a consequence of the disruption of the intestinal barrier [40].

In line with previous studies of immunoproteasome inhibition [7,9,41], body weight loss and the shortening of colon length were improved by ONX 0914 treatment, and the frequency of Th17 cells was strongly reduced when ONX 0914 was administered starting from the beginning. Previous studies using immunoproteasome inhibitors have so far only investigated Th17 cells at the peak of the colitis symptoms. For example, Kalim et al. showed that the daily application of ONX 0914 reduced the frequency of Th17 cells in the lamina propria of DSS mice on day 9 [7]. Moreover, other studies analyzing IL-17 expression levels and IL-17 secretion also reported reductions upon immunoproteasome inhibition and also in LMP7-KO mice, indicating that the Th17 response is indeed impaired [6,9]. In order to assess whether the development of Th17 cells is directly affected by immunoproteasome inhibition, different time points were analyzed. In line with the normal kinetics of the adaptive immune response, the frequency of Th17 cells increased in the lamina propria from day 6 to day 8 in vehicle-treated mice. In contrast, the frequency of Th17 cells stayed low on both days in the ONX 0914 group, indicating that Th17 cells did not develop in the first place.

Tregs are crucial for the regulation of immune responses by limiting excessive inflammation and subsequent tissue damage, as well as for preventing autoimmunity. Previous data regarding the influence of immunoproteasome inhibition on the differentiation of Tregs in vitro yielded opposing results. While Kalim et al. demonstrated enhanced Treg differentiation in the presence of ONX 0914 [7], Schmidt et al. did not observe a difference between vehicle and ONX 0914 groups [6]. Since in vitro differentiated Tregs are reported to have different functions than the ones in vivo [42] and thus might not reflect the influence of ONX 0914 in physiological conditions, Tregs were investigated in DSS-induced colitis, where they are strongly induced [43]. Indeed, in the lamina propria of DSS-treated mice, the frequency of Tregs was increased on day 6 and day 8, but no differences between vehicle and ONX 0914 treated mice were observed. Interestingly, the previous report that Tregs were more abundant in ONX 0914 mice during DSS-induced colitis [8] did not investigate the lamina propria on day 8 (where and when Th17 cells were analyzed), but the mesenteric lymph node on day 6. The analysis of the lymph node in the current study did not reveal differences between the DSS groups and naïve mice on day 6 (Appendix A) but only on day 8, which is in line with the reported time frame for the development of Tregs in DSS-induced colitis [43]. Moreover, the decrease in Tregs in the lymph node on day 8 detected upon ONX 0914 treatment could result from generally reduced inflammation. It was previously shown that immunoproteasome inhibition impairs the IL-2 secretion of activated T cells [6]. Since Tregs require strong IL-2 signaling [44], reduced IL-2 levels could also result in lower levels of Tregs. With the current setup, it is not possible to determine whether the induction of Tregs is impaired on a cell-intrinsic level or if it is a consequence of decreased inflammation and decreased IL-2 levels. Because Tregs could readily be identified in the lamina propria, the first hypothesis seems rather unlikely. Taken together, it can be concluded that Treg differentiation is most likely not impaired by immunoproteasome inhibition, and no evidence was found that it enhances their differentiation as proposed before. Nevertheless, functional characteristics of Tregs were not assessed in this study, and thus, no conclusion can be drawn about the effect of immunoproteasome inhibition on their suppressive capacity.

It is important to note that the inflammation induced by DSS is not mediated by the adaptive but by the innate immune response [45], and thus, the induction of Th17 cells is rather a consequence of the inflammation. Accordingly, the impaired function of the innate immune response could result in decreased Th17 cell formation. The establishment of the Th17 response in the intestine requires antigen presentation and co-stimulation by APCs such as DCs [34]. Therefore, an impaired function of APCs could lead to the observed reduced Th17 response. Moreover, it was previously shown that the immunoproteasome inhibition of either CD4^+^ T cells or APCs impaired the *Candida albicans*-induced polarization of Th17 cells, which was dependent on MHC-II function [46]. In animal models of colitis, the depletion of DCs has produced contradicting results, with the protection [47] or exacerbation [48] of the disease. However, it seems that DCs are crucial in producing cytokines to activate and maintain Th17 cells in experimental colitis [49,50,51,52,53]. The analysis of the CD11c^+^ population in our study demonstrated a slight reduction in their abundance in ONX 0914-treated mice on day 6 and day 8, suggesting that a reduced activation of T cells by DCs might partially play a role in the observed decrease in Th17 cells. But since the detected differences were not very strong, we assume this is not the driving factor for the observed differences. Nevertheless, further analyses, and more functional ones, would be required to definitively address this possibility; however, a detailed analysis of innate immune cells was not part of the current study.

Another possible explanation for the reduction in Th17 cells could be the impaired activation of T cells, since proper activation is critically required for their differentiation. Indeed, similar to the literature [6,19,54,55], we could measure the reduced expression of the activation marker CD44 in the spleens of ONX 0914 treated mice, already at early stages of colitis. In line with this, expression of several activation markers was strongly reduced by immunoproteasome inhibition in vitro, while retention in a naïve phenotype could not be observed, since CD62L, a marker for naïve cells, was strongly downregulated. Although immunoproteasome inhibition barely influenced the vitality of early activated T cells (12 h), cell death could be observed after 24 h post stimulation. This indicates that the initial activation of T cells occurs in immunoproteasome-inhibited T cells; however, the full activation of T cells cannot be accomplished in the presence of immunoproteasome inhibition. It seems that the improper activation of T cells in the presence of immunoproteasome inhibition leads finally to the death of some T cells. However, the fate of the residual T cells, and particularly in vivo activated T cells, remains to be investigated. It was suggested that immunoproteasome inhibition impairs T cell activation by restraining ERK signaling and proteostasis [6]. Already-differentiated Th17 cells are not affected by immunoproteasome inhibition when there is no strong activation trigger present. Hence, it seems that full proteasome capacity is particularly required during the early activation of T cells to maintain protein homeostasis. Immunoproteasome inhibition might affect IFN signaling, which could affect the broader inflammatory response, including the suppression of key cytokines such as IL-6 and IL-23, which are essential for Th17 differentiation and survival. Even though, to our knowledge, no other study has investigated the survival of differentiated Th17 cells in colitis, the survival of Th17 cells upon immunoproteasome inhibition was assessed in a therapeutic setting of EAE. It was shown that the therapeutic application of ONX 0914 after the establishment of active EAE could also ameliorate disease symptoms and decrease the Th17 response. Moreover, the adoptive transfer of in vitro stimulated PLP_139-151_-reactive T cells led to the induction of EAE in vehicle- but not ONX 0914-treated mice, indicating that the observed effect was mediated by the impaired function of antigen-specific T cells [10]. Indeed, immunoproteasome inhibition has been shown to reduce the secretion of pro-inflammatory cytokines [7,8,10,19].

## 5. Study Limitations

In our study, we used two different animal models (DSS-induced colitis and HDM-induced airway inflammation) to investigate the effect of immunoproteasome inhibition on Th17 survival. Whether similar results can be obtained in other models for Th17 differentiation remains to be determined. However, in both animal models we can observe a solid and robust Th17 population. Therefore, we strongly assume that our results can be generalized and reflect Th17 differentiation and survival.

The analysis of T cell activation in our study was limited to a few activation markers. To get a more comprehensive analysis of the effect of immunoproteasome inhibition on Th17 cells and to identify potential signaling pathways, it would be interesting to expand the analysis by performing RNA-Seq to profile the transcriptomes of mesenteric lymph nodes and the whole colon of ONX 0914-treated mice. Furthermore, investigations into IL-17 expression levels, which we are technically not able to perform, can be addressed in such an experiment.

Different studies reported a reduction of Th17 cells upon immunoproteasome inhibition using various inhibitors [7,8,9,10,11,12,13,56,57,58]. However, in our study, only ONX 0914 was applied to investigate the effect of immunoproteasome inhibition on Th17 differentiation and survival, which might limit the interpretation of our results. Nevertheless, it has been shown that dual immunoproteasome inhibition (LMP7/LMP2 or LMP7/MECL-1) is required to be effective in modifying cytokine secretion, the amelioration of autoimmunity, and Th17 differentiation [58,59]. Similar to the clinical candidate KZR-616 [59], ONX 0914 perfectly fulfills these requirements [8,58]. Hence, although our study is limited to the use of ONX 0914, this drug encompasses all the requirements of an effective immunoproteasome inhibitor and reflects the properties of the clinical candidate KZR-616.

## 6. Conclusions

Our data indicate that Th17 cell development is strongly impaired by immunoproteasome inhibition at an early phase, possibly by compromising proper activation, while already-differentiated Th17 cell were less susceptible to ONX 0914 in a delayed treatment regime. This knowledge about temporal aspects of the effect of immunoproteasome inhibition on Th17 cells is crucial for future therapeutic considerations. Therefore, our data not only provide insight into the functional role of the immunoproteasome in Th17 biology, but also emphasize the importance of carefully planning and optimizing treatment schedules accordingly in future clinical use of immunoproteasome inhibitors.

## Figures and Tables

**Figure 2 cells-14-00689-f002:**
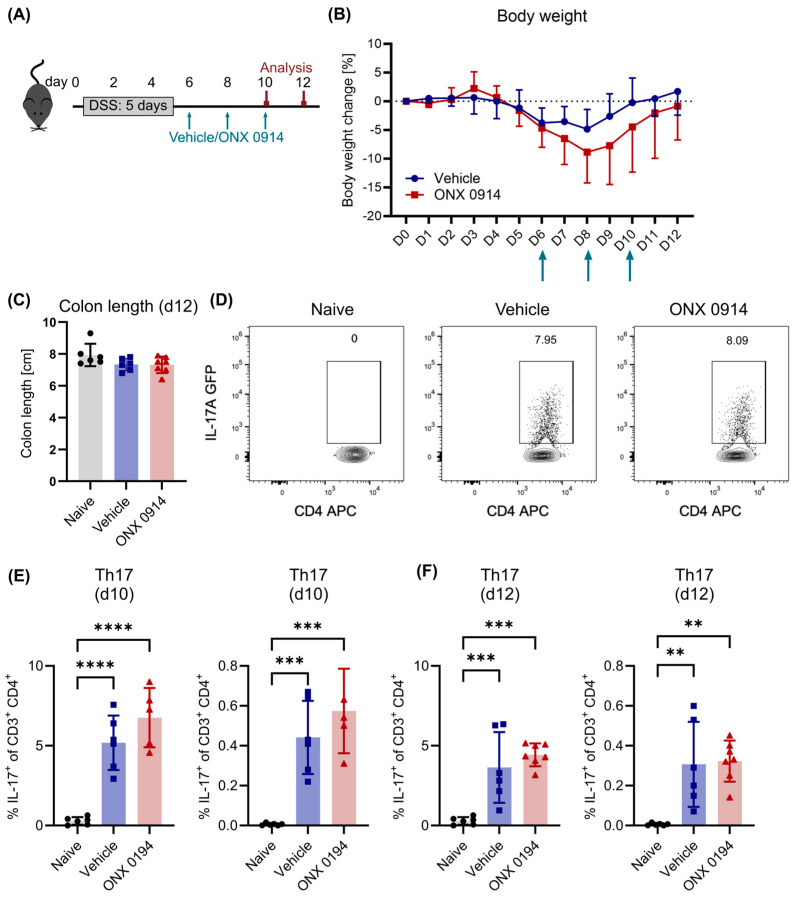
Similar recovery from DSS-induced colitis with or without immunoproteasome inhibition. Here, 2% DSS was added to the drinking water of IL-17A-GFP mice for 5 days, followed by normal drinking water. Starting on day 6, vehicle or ONX 0914 (10 mg/kg) was administered subcutaneously every other day and mice were sacrificed on (**D**,**E**) day 10 and (**C**,**F**) 12 as depicted in (**A**). Respective tissues were dissected and single cell suspensions were obtained by enzymatic and/or mechanic dissociation to allow further analysis by flow cytometry. (**B**) Body weight was monitored daily. Percent weight loss relative to the weight on day 0 (*y*-axis) is plotted versus time in days (*x*-axis). Treatment time points are indicated with arrows. (**C**) Mice were sacrificed on day 10 to measure the colon length (caudal of the appendix). (**D**) Gating scheme to analyze Th17 cells. After the exclusion of doublets and dead cells (gating strategy described in Appendix A), cells were identified as CD3^+^ CD4^+^ and subsequently analyzed for the expression of IL-17A-GFP. Graph shows a sample of the lamina propria on day 10. (**E**,**F**) Frequency of Th17 cells among CD3^+^ CD4^+^ T cells (left) and among all viable cells (right) on day 10 (**E**) and day 12 (**F**). Graphs show pooled data of 2 independent experiments (naïve—*n* = 6, vehicle/ONX 0914—*n* = 5–7). Data are shown as mean ± D (**B**) or ± SD (**C**–**F**). (**C**–**F**) Each point represents an individual mouse. ** *p* < 0.01, *** *p* < 0.001, **** *p* < 0.0001 (1- or 2-way ANOVA).

**Figure 3 cells-14-00689-f003:**
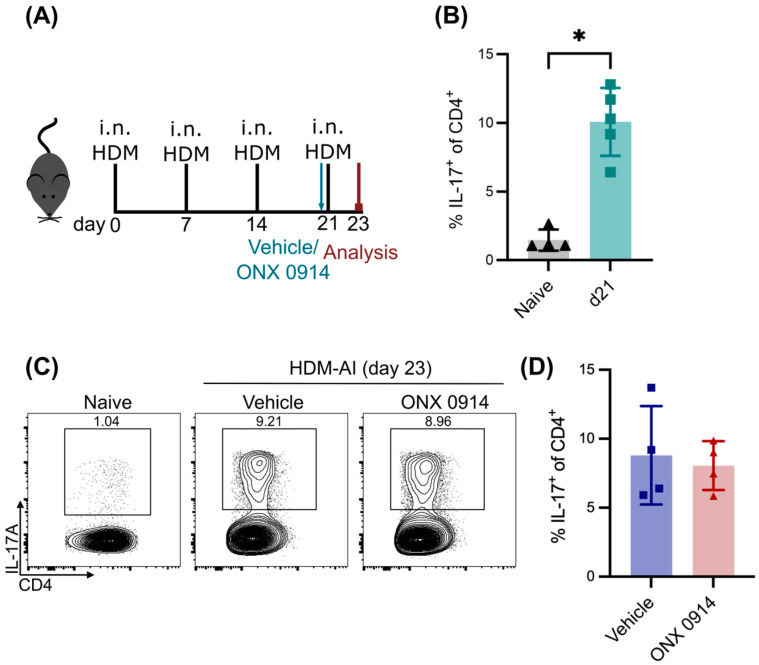
Th17 cells are not affected by immunoproteasome inhibition in house dust mite-induced airway inflammation (HDM-AI). (**A**) Experimental scheme. (**B**) GATIR mice received three intranasal applications of 50 µg HDM extract on days 0, 7 and 14. The Th17 response in the lung was analyzed on day 21 by flow cytometric analysis of the dissected and dissociated lung tissue. (**C**,**D**) GATIR mice received four intranasal applications of 50 µg HDM extract on day 0, 7, 14 and 21. One hour before the last immunization, mice were treated with 10 mg/kg ONX 0914 or vehicle subcutaneously. Inflammatory infiltration was analyzed by flow cytometry on day 23 by flow cytometric analysis of the dissected and dissociated lung tissue. Naïve mice served as controls. For all experiments, samples were restimulated in vitro for 5 h with PMA/ionomycin and treated with brefeldin A to analyze cytokine expression. (**C**) Representative plots showing the Th17 cell population in the lung after exclusion of doublets and dead cells and gating on CD4^+^. (**D**) Frequency of the Th17 cell population among all CD4^+^ T cells. Each point represents an individual mouse. Data from day 23 are representative for two independent experiments (*n* = 4). Results from day 21 show n = 5, derived from one experiment. Data are shown as mean ± SD. * *p* < 0.01 (Mann–Whitney test).

**Figure 4 cells-14-00689-f004:**
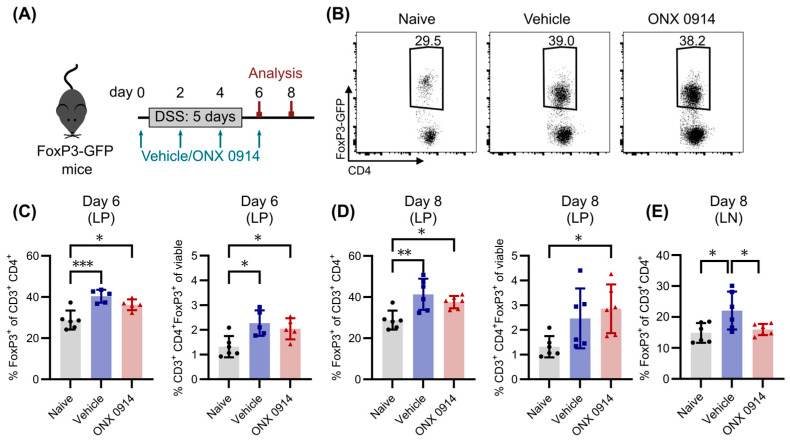
Immunoproteasome inhibition does not change the frequency of Tregs in the lamina propria during DSS-induced colitis. Here, 2% DSS was added to the drinking water of FoxP3-GFP mice for 5 days, followed by normal drinking water. Simultaneously, mice received subcutaneous injections of vehicle or ONX 0914 (10 mg/kg) every other day. Mice were sacrificed on day 6 or day 8 and respective tissues were dissected. Single cell suspensions were obtained by enzymatic and/or mechanic dissociation to allow further analysis by flow cytometry. (**A**) Experimental setup. (**B**) Representative FACS plots of FoxP3-GFP^+^ Tregs in the lamina propria (LP) after exclusion of doublets and dead cells and pre-gating on CD3^+^ CD4^+^. Graph shows a naïve sample and samples on day 8. (**C**,**D**) Frequency of Tregs cells among CD4^+^ T cells (left) and among all viable cells (right) on day 6 (**C**) and day 8 (**D**) in the lamina propria (LP). (**E**) Frequency of Tregs among CD4^+^ T cells in the mesenteric lymph node (LN) on day 8. Graphs show pooled data of two independent experiments. Data are shown as mean ± SD, and each point represents an individual mouse (*n* = 5–6). * *p* < 0.05, ** *p* < 0.01, *** *p* < 0.001 (1-way ANOVA).

**Figure 5 cells-14-00689-f005:**
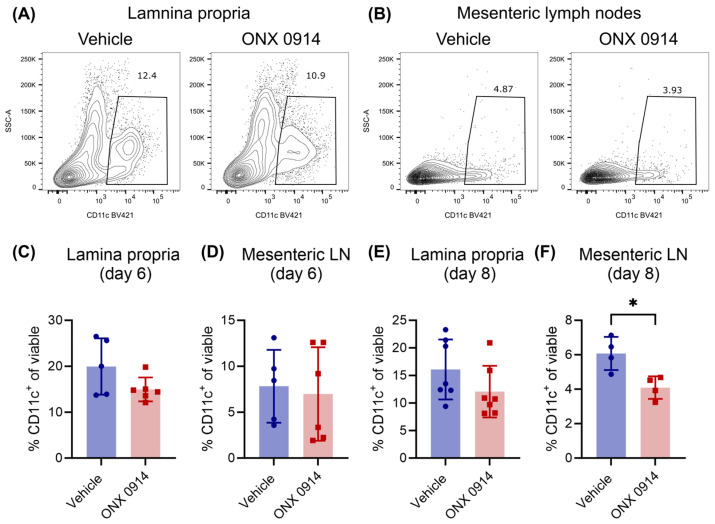
Immunoproteasome inhibition slightly reduces dendritic cells in mesenteric lymph nodes. Here, 2% DSS was added to the drinking water of IL-17A-GFP mice for 5 days, followed by normal drinking water afterwards. Simultaneously, mice received subcutaneous injections of vehicle or ONX 0914 (10 mg/kg) every other day starting on day 0. Mice were sacrificed on day 6 or day 8 and respective tissues were dissected. Single cell suspensions were obtained by enzymatic and/or mechanic dissociation to allow further analysis by flow cytometry. The frequency of CD11c^+^ among all viable cells (after exclusion of doublets and dead cells) was analyzed on (**C**) day 6 and (**A**,**B**,**E**,**F**) day 8 in the lamina propria (**A**,**C**,**E**) and mesenteric lymph nodes (**B**,**D**,**F**). Gating schemes are shown for the lamina propria (**A**) and the mesenteric lymph node (**B**) on day 8. Graphs show pooled data of 2–3 independent experiments (**n** = 4–7). Data are shown as mean ± SD, and each point represents an individual mouse. * *p* < 0.05 (unpaired, two-tailed student’s *t* test).

**Figure 6 cells-14-00689-f006:**
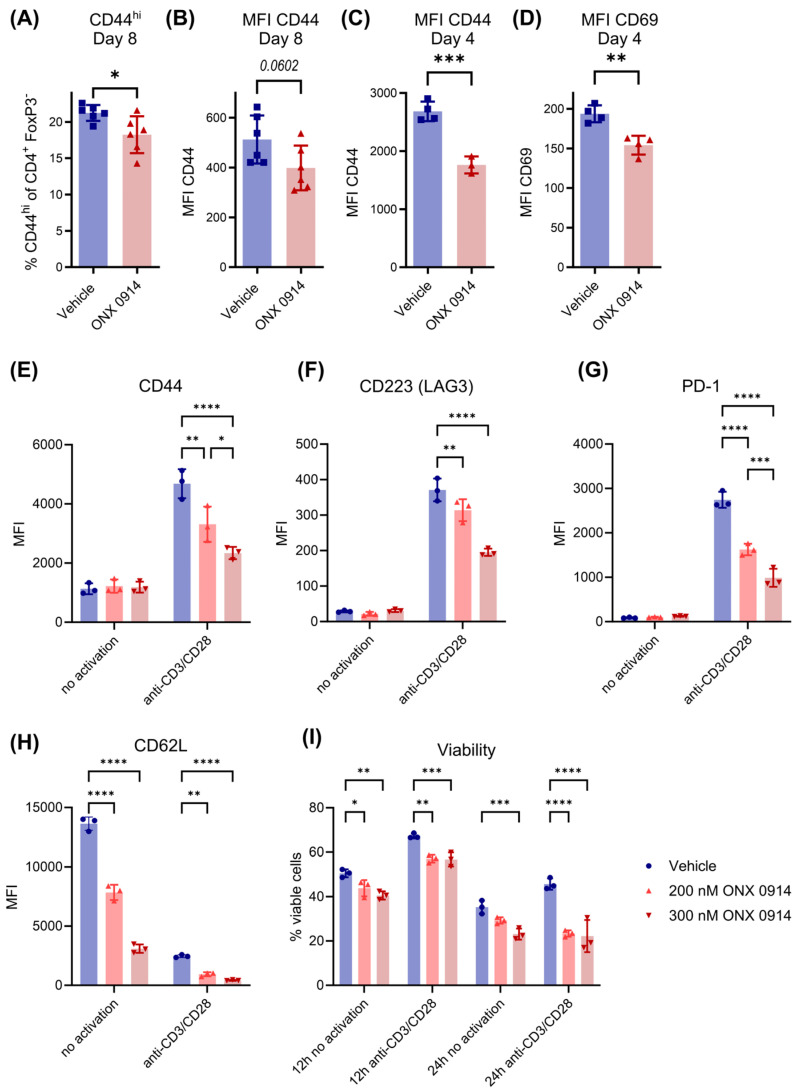
Impaired T cell activation by immunoproteasome inhibition. (**A**–**D**) Here, 2% DSS was added to the drinking water of (**A**,**B**) FoxP3-GFP and (**C**,**D**) IL-17A-GFP mice for 4–5 days followed by normal drinking water. Simultaneously, mice received subcutaneous injections of vehicle or ONX 0914 (10 mg/kg) every other day. (**A**,**B**) Mice were sacrificed on day 8 and CD44 expression was analyzed on non-Treg T cells identified as CD4^+^ FoxP3^-^ cells in the spleen after exclusion of dead cells and doublets. (**A**) Frequency of the CD44^hi^ population among CD4^+^ FoxP3^-^ and (**B**) mean fluorescence intensity (MFI) of CD44 on CD4^+^ FoxP3^-^ (pooled data from two independent experiments, n = 6). (**C**,**D**) Mice were sacrificed on day 4 and the expression of activation markers (**C**) CD44 and (**D**) CD69 was analyzed in the spleen on CD4^+^ T cells (*n* = 3–4). (**E**–**I**) CD4^+^ T cells were magnetically isolated from spleens of naïve mice and activated in vitro with plate-bound anti-CD3/CD28 antibodies. Samples were simultaneously treated with different concentrations of ONX 0914 or vehicle (DMSO) as control. (**E**–**H**) Expressions of different surface activation markers were analyzed after 24 h by flow cytometry (*n* = 3, representative of independent experiments yielding similar results). (**I**) Viability was analyzed after 12 and 24 h by flow cytometry (*n* = 3, representative of independent experiments yielding similar results). Data are shown as mean ± SD, and each point per group represents an individual mouse. * *p* < 0.05, ** *p* < 0.01, *** *p* < 0.001, **** *p* < 0.0001 (unpaired, two-tailed student’s *t* test or 2-way ANOVA).

## Data Availability

The raw data supporting the conclusions of this article will be made available by the authors on request.

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
