# Peer review of "Immunoproteasome Inhibition Impairs Differentiation but Not Survival of T Helper 17 Cells"

_cells, 2025, doi:10.3390/cells14100689_

Round 1
Reviewer 1 Report
Comments and Suggestions for Authors In this manuscript, Oliveri et al. propose that immunoproteasome inhibition impaired Th17 cell differentiation but not their survival. It had been previously demonstrated that immunoproteasome inhibition ameliorated disease phenotypes in autoimmune diseases in mice models such as colitis, rheumatoid arthritis, SLE, etc. The amelioration of disease phenotypes in these several disease models was attributed to the reduction of pro-inflammatory Th17 cells. This study looked at two different disease models in mice (DSS-induced colitis and HDM-induced asthma) to address whether immunoproteasome inhibition impaired the activation of T cells, including Th17 differentiation, or its survival. The first part of the manuscript recapitulated findings from previous studies showing that simultaneous timing of immunoproteasome inhibition and induction of DSS-induced colitis ameliorated disease phenotypes and reduced pro-inflammatory Th17 cells. The authors then showed that this reduction in Th17 cells was not caused by impaired survival due to immunoproteasome inhibition in both disease models. Furthermore, the authors showed that Tregs were not affected by immunoproteasome inhibition. They report a modest reduction of DCs on day 8, which might contribute to reduced Th17 cells. However, the authors concluded that the size effect was too small to be the driving force behind the reduction of Th17. Furthermore, the authors showed that immunoproteasome inhibition did not seem to block the transition from the naïve to the activated state of T cells but rather reduced the expression of activation markers. Lastly, immunoproteasome inhibition seemed to only modestly affect the viability of T cells after 12h. After 24h, activated T-cells lose significant viability, probably due to increased demand of proteasome activity upon activating stimuli. Th17 cells were not especially more sensitive compared to IL17- Th cells. Immunoproteasome inhibition is a relatively unexplored therapeutical avenue. Numerous studies have shown promising data on the use of specific immunoproteasome inhibition for several diseases related to the immune system, including cancer. This study contributes to a better understanding of how immunoproteasome affects T helper cells, specifically in autoimmune diseases. This is an interesting study. The data are convincing. I have only a few suggestions: 1. In Figure 5F, the data shows a decrease of DCs in the LN in the presence of ONX. The possibility that DCs may contribute to reduced Th17 is not investigated in detail. While this is mentioned in the discussion, this point could be further discussed, and the authors could test the possibility of interrogating the DC contribution of TH17 cells in their models. 2. To get a more comprehensive analysis of immunoproteasome contribution, it would be interesting to expand the marker analysis and do a systematic screening to see which specific activation markers CD4+ cells are affected and which ones are not affected in the presence of ONX. This may help address specific mechanisms versus broader population effects Minor points 1. Figure 6A-6D; to clarify, I would recommend showing on the figure which data belongs to which day (as in previous figures)-Author Response
Please see the attachment.

Reviewer 2 Report
Comments and Suggestions for Authors
Summary: The immunoproteasome (iCP) has been envisioned as a target for monocytic and T helper cell mediated diseases because of its role in the activation, differentiation and function of T cell subsets. Here, Authors employed mouse models of DSS-induced colitis and airway inflammation to ask if an immunoproteasome inhibition altered Th17 generation or altered the Treg-to-Th17 cells ratio over time.
Initial studies showed elevated numbers of CD4+ IL-17+ T cell accumulated in both DSS and HDM-AI models (Figs 1-3). In the DSS model, prophylactic treatment with iCP inhibitor greatly reduced disease and CD4+Th17+ T cell accumulation. Repeating this exp design but giving iCP later during recovery period did not alter accumulation of CD4+Th17 cells in LPL. Authors also evaluated the effect of iCP treatment on Treg accumulation in DSS, and found no differences vs vehicle. Further analyses of DC in the DSS model suggested iCP does not alter the accumulation of DCs, however, the data were too variable.
In the HDM model, CD4+ Th17+ cells had increased in lungs at 21 days. Treatment with iCP at 21 days and performing analysis 2 days later, revealed no effect on CD4+Th17+ T cell accumulation. Likewise testing for an effect of iCP on T regs in the HDM model revealed no effect. Subsequent studies (Fig 6/Sup fig 6)) examined whether iCP treatment influenced CD$+ T cell activation in the DSS model. These studies are a good starting point but are not complete and Figure 6 legend and text are confusing.
Comments and suggestions for Authors:
Major:
- Methods: Authors should provide rationale/data why 10 mg/kg of the inhibitor was used in this study.
- Fig 6 in vivo studies with the iCP and DSS are confusing and not well explained in legend or text. First, the legend for Fig 6 A-D states IL-17A-GFP and FOXP3-GFP mice were subjected to DSS and treated with vehicle/iCP, however, no data were shown for FOXP3+ or for IL-17GFP+ cells. Further confusing is that the text states only FOXP3-GFP+ mice were used (Pg 7, lines 335 -337). Presenting data for FOXP3- T cells seem disconnected to study. Please clarify.
Second, most relevant if Authors also measured the actual amount of IL-17 produced in these cells in experiments shown in Fig 6. While state of T cell activation is important, determining levels of cytokines produced would be more informative.
- The data in Sup Fig 6A is incomplete. The data raise the question of what is the appropriate concentration of ONX 0914 for in vitro assays that blocks iCP in IL17+/- CD4 T cells but does not initiate significant apoptosis?
Minor:
- Methods: Pg 3, line 101, Authors should explain/give rationale why in vivo studies solubilized the iCP inhibitor in 10 % sulfobutylether-β-cyclodex-101 trin (w/v; Captisol) and 10 mM sodium citrate (pH 6) whereas they used a different solvent, DMSO, for in vitro studies (pg 4, line 111). Why was captisol not used in the in vitro too?
- Methods included mention of GATIR (Gata3tm1.1Hjf) [30] mice, but I did not see any data employing these mice. Please clarify.
Reviewer 3 Report
Comments and Suggestions for Authors
In this manuscript, Oliveri and colleagues present an impaired differentiation of T helper 17 cells upon immunoproteasome inhibition by the drug ONX 0914. The authors make use of different in vivo and in vitro models to study the effect of immunoproteasome inhibition. In the past, this group and others could show that the inhibition of the immunoproteasome mainly affects Th17 cells. Up to now, it remained elusive whether or not the differentiation or survival of Th17 cells are affected.
The authors use DSS-induced colitis and house dust mice airway inflammation mouse models to show that Th17 differentiation and Treg cells are not affected by immunoproteasome inhibition. Interestingly, application of ONX 0914 lead to an impaired development of Th17 cells. On the other hand, differentiated Th17 cells are unaffected by inhibition of the immunoproteasome.
Furthermore, the authors describe the effects of immunoproteasome inhibition for dendritic cells in a DSS-induced colitis model. Lastly, the group observe a diminished T cell activation in vivo and in vitro upon immunoproteasome inhibition.
Overall, the manuscript provides an overview of consequences of immunoproteasome inhibition by ONX 0914 for Th17 and Treg cells, dendritic cells and gives insight about an altered T cell activation.
Criticism in detail:
Why did the authors focus on ONX 0914 as immunoproteasome inhibitor? Did they use other imhibitors, e.g. the FDA-approved bortezomib or carfilzomib? If so, could the authors reproduce these findings?
Line 196-210: This section still includes advices about how to write this part.
Line 471-474: This section still includes advices about how to write this part.
Material and Methods:
- Why did the authors use different ways of reconstitution of ONX0914 for in vivo and in vitro use?
- Why did the authors measure the intracellular cytokine staining exclusively on the LSRFortessa?
Results:
Figure 1: Please add ‘lamina propria’ in E and F. If D is a representative gating scheme, the indicated frequencies should be found in F, right? In F are no respective values for 1,16/7,57 or 2,10.
The authors show a reduced frequency of Th17 cells upon immunoproteasome inhibition. Can they comment on absolute numbers? What about the survival of these cells? The authors discuss that this finding was not due to a general decrease of CD4+ T cells, but what about Treg cells here? The question if survival of Th17 cells is affected remains open since no experiments in that direction were performed.
Figure 2: In 2D, the authors present the same naïve control plot for day 10 as in Fig. 1D for day 8.
Figure 3: The authors present a therapeutic application of ONX 0914 for the HDM-AI disease model. Can they please comment on a “prophylactic” setting for HDM-AI as presented in Figure 1 for the DSS-induced colitis?
Why do the authors refer to Figure S3 (Treg gating strategy) in line 281? Tregs are part of
Figure 4.
At the end of this figure, the authors conclude that ONX 0914 application prevents Th17 development – this was not shown for the HDM-AI model. Please correct this sentence accordingly or explain this conclusion in more detail. From my perspective, this conclusion is only true for Figure 1. Line 303/304: Please include the data (freq. of Treg cells upon ONX 0914 treatment after DSS treatment).
Figure 6: Why did the authors not apply the apoptosis analysis ex vivo? Another idea would have been to sort for Th17 cells in the DSS-colitis model at day 8 and perform (Figure 1). In addition to AnnexinV, cleaved caspase 3 staining/7-AAD would have been of interest. Please comment on this.
Supplementary Figures:
- Please add frequencies of the indicated gates in Figure S1 and S3.
Abstract:
It would be beneficial to include a sentence about the findings on DCs.

Round 2
Reviewer 2 Report
Comments and Suggestions for Authors
Revisions improved the report but sill average priority score.
Author Response
No specific comment from the reviewer, which we have to address.
Reviewer 3 Report
Comments and Suggestions for Authors
No further comments or suggestions. Thanks for addressing my comments.
Author Response

(The authors gave the same response as above.)
